# Exploring Plant-Based Compounds as Alternatives for Targeting *Enterococcus faecalis* in Endodontic Therapy: A Molecular Docking Approach

**DOI:** 10.3390/ijms25147727

**Published:** 2024-07-15

**Authors:** Nezar Boreak, Rahf Zuhair Al Mahde, Waseem Ahmed Otayn, Amwaj Yahya Alamer, Taif Alrajhi, Shatha Jafri, Amnah Sharwani, Entesar Swaidi, Shahad Abozoah, Ahlam Abdu Mohammed Mowkly

**Affiliations:** 1Department of Restorative Dental Sciences, College of Dentistry, Jazan University, Jazan 45142, Saudi Arabia; rahafb1421@gmail.com (R.Z.A.M.); alameramwaj@gmail.com (A.Y.A.); alrajhiteef@gmail.com (T.A.); dr.shatha2000@gmail.com (S.J.); amnahsher@gmail.com (A.S.);; 2Specialized Dental Canter, Ministry of Health, Jazan 45142, Saudi Arabia; waseema055838@gmail.com (W.A.O.);

**Keywords:** endodontic infections, *Enterococcus faecalis*, SrtA, plant-based compounds, molecular docking, alternative treatments

## Abstract

Endodontic infections pose significant challenges in dental practice due to their persistence and potential complications. Among the causative agents, *Enterococcus faecalis* stands out for its ability to form biofilms and develop resistance to conventional antibiotics, leading to treatment failures and recurrent infections. The urgent need for alternative treatments arises from the growing concern over antibiotic resistance and the limitations of current therapeutic options in combating *E. faecalis*-associated endodontic infections. Plant-based natural compounds offer a promising avenue for exploration, given their diverse bioactive properties and potential as sources of novel antimicrobial agents. In this study, molecular docking and dynamics simulations are employed to explore the interactions between SrtA, a key enzyme in *E. faecalis*, and plant-based natural compounds. Analysis of phytocompounds through molecular docking unveiled several candidates with binding energies surpassing that of the control drug, ampicillin, with pinocembrin emerging as the lead compound due to its strong interactions with key residues of SrtA. Comparative analysis with ampicillin underscored varying degrees of structural similarity among the study compounds. Molecular dynamics simulations provided deeper insights into the dynamic behavior and stability of protein–ligand complexes, with pinocembrin demonstrating minimal conformational changes and effective stabilization of the N-terminal region. Free energy landscape analysis supported pinocembrin’s stabilizing effects, further corroborated by hydrogen bond analysis. Additionally, physicochemical properties analysis highlighted the drug-likeness of pinocembrin and glabridin. Overall, this study elucidates the potential anti-bacterial properties of selected phytocompounds against *E. faecalis* infections, with pinocembrin emerging as a promising lead compound for further drug development efforts, offering new avenues for combating bacterial infections and advancing therapeutic interventions in endodontic practice.

## 1. Introduction

*Enterococcus faecalis* (*E. faecalis*) is a Gram-positive bacterium frequently associated with persistent endodontic infections. Although it constitutes a minor portion of the flora in untreated canals, it plays a significant role in the development of persistent periradicular lesions following root canal treatment [1,2]. *E. faecalis* is often found in a high percentage of root canal failures and can persist in the root canal either as a solitary organism or as a dominant component of the microbial community [3].

*E. faecalis* is equipped with various virulence factors and can share these traits with other species in the oral cavity, enhancing its survival and pathogenicity [4]. Despite relying less on these virulence factors, *E. faecalis* excels in its ability to persist and thrive in the root canals of teeth [5]. It has been demonstrated to withstand intracanal dressings of calcium hydroxide when residing in dentinal tubules [6,7]. Additionally, *E. faecalis* can form biofilms, which significantly increase its resistance to destruction, making it more resilient against phagocytosis, antibodies, and antimicrobials compared to non-biofilm-producing organisms [8].

Sortases are extracellular transpeptidases located in the plasma membrane of Gram-positive bacteria. These enzymes sort and anchor proteins to the cell surface by cleaving the conserved threonine in C-terminal LPXTG-like motifs and forming an amide bond between the threonine and the pentaglycine cross-bridge of cell wall peptidoglycan [9,10,11]. Genome analysis of *E. faecalis* has identified major sortases, including class A sortase (SrtA), which functions as a housekeeping sortase, and class C sortase (SrtC), which participates in pilus polymerization [12].

In *E. faecalis*, sortase enzymes facilitate the cell wall anchoring of virulence factors through a three-step process [10]. First, SrtA recognizes its substrate protein by the conserved LPXTG motif, preparing it for proteolytic cleavage. These substrate proteins (ex., endocarditis and biofilm-associated pili proteins—EbpA, EbpB, and EpbC) contain an LPXTG motif at the C-terminus, known as the Cell Wall Sorting Signal (CWSS), which encloses a pentapeptide LPXTG motif, a hydrophobic domain, and a positively charged tail [10]. SrtA halts the proteins at the Sec channel to finalize the sorting process [13]. In the second step, SrtA cleaves the proteins between the threonine (T) and glycine (G) residues of the LPXTG motif, forming a thioester acyl-enzyme intermediate. This intermediate then attaches to the peptidoglycan precursor, Lipid II, through the threonine residue. The third step involves nucleophilic attack and covalent anchoring. The intermediate complex is resolved by the attack of an amine nucleophile from the pentaglycine-branched Lipid II, covalently linking the Lipid II-coupled protein to the cell wall peptidoglycan and releasing SrtA for further sorting [14,15]. SrtC then polymerizes the SrtA-processed substrates, such as endocarditis and biofilm-associated pili proteins.

SrtA has been shown to be essential for pilus assembly [16], biofilm formation [12], and host tissue colonization [17]. In addition, its non-essential role in Gram-positive bacterial growth or viability and its convenient location on the cell membrane, which makes it more accessible to inhibitors, further support the notion that SrtA is an ideal target for anti-virulence drug development. Consequently, molecules that inhibit SrtA activity are regarded as promising antimicrobial compounds [18].

Antimicrobial drugs have demonstrated remarkable efficacy in managing bacterial infections [19]. However, bacterial pathogens are unlikely to surrender entirely to treatment, as they have developed multiple defense mechanisms against antimicrobials over time. There is an urgent need for novel antibiotics to combat this resistance. Organic compounds with anti-bacterial properties have recently gained attention for their potential in infection treatment [20]. Many phytocompounds exhibit low toxicity in mammals and are extensively used in treating various diseases [21,22]. Numerous phytocompounds also exhibit anti-bacterial activity, including against *E. faecalis* [23].

Computational studies present distinct advantages in the examination of the SrtA protein of *E. faecalis*, offering detailed insights into atomic-level molecular interactions and binding mechanisms. This capability enables efficient screening of a wide array of chemical compounds for potential inhibitors. Computational approaches are cost-effective, expedient, and capable of exploring a broader spectrum of chemical diversity, thereby aiding in the discovery of new lead compounds with therapeutic promise across various fields, including endodontics [24,25,26].

In this study, we explore the binding affinity of thirty natural phytocompounds to the SrtA protein through computational analysis. Molecular docking was employed to assess the binding affinity of these phytocompounds to SrtA. Additionally, molecular dynamics (MD) simulations were performed to evaluate the stability of the docked poses of the top-ranked inhibitors.

## 2. Results

### 2.1. SrtA Protein Structure

The crucial step in protein structure modeling involves identifying a template protein structure with high query coverage, sequence identity, and low E-value. SrtA of Bacillus anthracis was selected as a suitable template based on these criteria, featuring 61% query coverage, an E-value of 6 × 10^−19^, and 33.78% sequence identity. The top three models predicted by Modeller were chosen for structural validation. Despite undergoing loop refinement, the first eighty residues of the modeled protein formed a loop region. This may lead to high fluctuation in MDS analysis. Consequently, these loop regions were excluded from further analysis. The selected models were validated for quality using a Ramachandran plot, ERRAT, and ProSA (Table 1). Additionally, the root mean square deviation (RMSD) between the modeled and template structures was below 2 Å for all three top models. Among the predicted models, model one demonstrated good conformational stability and was deemed suitable for molecular docking and dynamics simulation studies (Figure 1). Energy minimization was performed using the CHARMM27 force field in GROMACS 2024.2 for modeled protein and for further analysis.

### 2.2. Molecular Docking Analysis

The molecular docking of thirty compounds with the SrtA protein generated the top ten poses for each compound. The optimal pose of the control compound (ampicillin), with a binding energy of −6.5 kcal/mol, was utilized as the threshold binding score for subsequent compound screening. Compounds exhibiting stronger binding affinity than the control compound were considered potential binders. Approximately twenty-four compounds demonstrated higher binding energies than the control drug, ranging from −8.0 to −6.8 kcal/mol (as shown in Table 2). The interacting residues of the SrtA protein with all the compounds were further scrutinized. In this study, the phytocompounds demonstrated diverse interaction profiles with the SrtA protein. The phytocompounds preferred forming hydrophobic interactions over hydrogen bonds. LYS230 predominantly mediated these hydrophobic interactions in approximately nine different ligands, while ALA92, VAL147, GLN227, LEU212, LEU91, THR209, and ARG234 facilitated these interactions in seven different ligands each. Conversely, ARG234 formed the most hydrogen bonds, participating in interactions with eight different ligands, followed by GLN227, GLU95, and ASP135, which contributed to interactions with six and five different ligands, respectively. 

The top two compounds with the highest binding affinity, pinocembrin and glabridin, were investigated further to understand the interactions (Figure 2). Pinocembrin, with a binding energy of −8.0 kcal/mol, led the table and favored hydrophobic interactions with ALA92, VAL147, LEU148, LEU212, and ILE223, and formed hydrogen bonds with VAL88 and CYS225. Glabridin, with binding energies of −7.9 kcal/mol, showed unique patterns; glabridin engaged LEU91, ALA92, and PHE137 hydrophobically, forming hydrogen bonds with VAL88, ALA92, THR224, and ARG234. The control drug, ampicillin, with a binding energy of −6.5 kcal/mol, primarily showed hydrophobic interactions with LEU91 and ALA92, hydrogen bonds with VAL88 and MET206, and salt bridge with ARG234. These findings emphasize the importance of hydrophobic interactions in ligand binding, with residues like LYS230 and ARG234 playing crucial roles across multiple interaction types. 

Furthermore, each molecule has undergone comparison with the control, revealing varying degrees of Tanimoto similarity. The summary of similarity scores between the study compounds and the control is depicted in Figure 3. These scores denote differing levels of similarity, ranging from 0.25 to 0.51, indicating that these molecules exhibit modest similarity to the control based on the Tanimoto similarity metrics. Notably, the betulinic acid derivative demonstrated the highest similarity (0.51) to ampicillin among the compounds, suggesting a greater overlap in structural features or characteristics with ampicillin compared to others. Following this, taxifolin showed a Tanimoto similarity of approximately 0.36, indicating a moderate level of similarity with ampicillin. A higher Tanimoto score, closer to 1, signifies greater similarity between molecules. However, the scores observed suggest that they likely do not share the same structural framework with the control drug.

### 2.3. Molecular Dynamics and Simulation Analysis

The MD simulations were employed to optimize and elucidate the potential binding mechanism of phytocompounds with the SrtA protein. The apoprotein and protein–ligand complexes involving pinocembrin, glabridin, and ampicillin underwent MDS analysis.

#### 2.3.1. Structural Stability Analysis

The fluctuation equilibrium and structural stability of the apoprotein and complex systems during MD simulations were examined by monitoring the backbone root mean square standard deviation (RMSD) with respect to their starting structures as a function of simulation time. The root mean square deviation (RMSD) values of the protein backbone were calculated over the course of 100 ns molecular dynamics (MD) simulations for four different systems: the apoprotein, the control drug–protein complex, the glabridin–protein complex, and the pinocembrin–protein complex. The resulting RMSD trajectories are presented in Figure 4a. The apoprotein showed an initial RMSD increase and stabilized around 0.4 nm, suggesting that the apoprotein undergoes significant conformational changes early in the simulation before reaching a relatively stable state. The fluctuations observed throughout the simulation indicate the inherent flexibility of the protein in the absence of any ligands. The control drug–protein complex exhibited a higher RMSD overall compared to the apoprotein, stabilizing around 0.5–0.6 nm, indicating more substantial conformational changes and flexibility within the protein structure. The increased RMSD and fluctuations suggest that the control drug affects the protein’s structural stability by interacting with key residues and altering the protein’s dynamic behavior. The glabridin–protein complex showed a similar trend to the control drug–protein complex but with slightly lower RMSD values, stabilizing around 0.4–0.5 nm. This suggests that glabridin binds to the protein that induces significant conformational changes, though slightly less pronounced than those induced by the control drug. The consistent RMSD values indicate that the glabridin–protein complex reaches a stable conformational state during the simulation. The pinocembrin–protein complex exhibited the lowest RMSD values among the complexes, stabilizing around 0.2–0.3 nm. This indicates that pinocembrin binding induces the least conformational change in the protein, suggesting a more stabilizing interaction compared to the other compounds. The lower RMSD values and reduced fluctuations imply that pinocembrin maintains the structural integrity of the protein more effectively through stabilizing interactions.

#### 2.3.2. Structural Flexibility Analysis

To evaluate and compare the structural flexibility between apoprotein and complex systems, the root mean square fluctuation (RMSF) values are computed from MD trajectories. The RMSF profiles are presented in Figure 4b. The RMSF values for the apoprotein indicated moderate fluctuations, particularly in the N-terminal region (residues 1–100), where fluctuations peak at approximately 1.2 nm, indicating apoprotein’s flexibility localized primarily in this region, while the remainder of the proteins exhibit lower fluctuations (0.2–0.5 nm), indicating a more stable structure. The control drug–protein complex showed increased RMSF values in the N-terminal region compared to the apoprotein, with fluctuations peaking around 1.5 nm, suggesting that the binding of the control drug enhances the flexibility of this region. The central and C-terminal regions exhibited similar fluctuation patterns to the apoprotein, indicating that the control drug primarily affects the N-terminal dynamics. The glabridin–protein complex exhibited an RMSF profile with fluctuations in the N-terminal region peaking around 1.1 nm, which is slightly lower than the control drug–protein complex. This indicated that glabridin induces less flexibility in the N-terminal region. The central and C-terminal regions showed similar fluctuations to the apoprotein and control drug–protein complex, suggesting a consistent dynamic behavior across these regions. The pinocembrin–protein complex showed the lowest RMSF values among the complexes, particularly in the N-terminal region, with peaks around 0.9 nm, indicating pinocembrin binding stabilizes the N-terminal region more effectively than the other compounds. The central and C-terminal regions displayed similar low fluctuations (0.2–0.4 nm), indicating that pinocembrin maintains the overall stability of the protein structure.

#### 2.3.3. Compactness of the Protein

The structural compactness of each system was analyzed by estimating the radius of gyration (Rg) from their respective MD trajectories (Figure 4c). The apoprotein showed a stable Rg value fluctuating around 1.5 nm, indicating that the apoprotein maintained a consistent level of compactness throughout the simulation, suggesting a stable structure in the absence of any ligands. The control drug–protein complex exhibited higher Rg values, fluctuating around 1.6–1.7 nm. This increase in Rg indicated that the control drug induces expansion in the protein structure, leading to a less compact and more dynamic protein conformation. The significant fluctuations in Rg suggested that the control drug affects the structural stability and induces conformational changes in the protein. The glabridin–protein complex showed Rg values similar to apoprotein, fluctuating around 1.5–1.6 nm, suggesting glabridin binding does not significantly alter the overall compactness of the protein compared to the apoprotein. The moderate fluctuations indicate that glabridin maintains the structural integrity of the protein while allowing for some flexibility. The pinocembrin–protein complex exhibited the lowest Rg values, maintaining a stable range around 1.5 nm. This indicated that pinocembrin binding resulted in a more compact and stable protein structure, with fewer fluctuations compared to the other complexes. The consistent Rg values suggest that pinocembrin stabilizes the protein, reducing its conformational flexibility.

#### 2.3.4. Hydrogen Bond Analysis

To understand the stability and interactions of the protein–ligand complexes, we analyzed the number of hydrogen bonds over the simulation time. The plot in Figure 4d depicts the number of hydrogen bonds formed between the protein and the ligands over a 100 ns simulation. The control drug–protein complex consistently maintained 1–3 hydrogen bonds throughout the simulation, indicating moderate stability. The variability in the number of hydrogen bonds suggested fluctuating interactions, which may contribute to the weaker binding affinity observed in the free energy analysis. The glabridin–protein complex showed a distinct pattern with 1–3 hydrogen bonds formed intermittently. The presence of hydrogen bonds was sporadic, suggesting fewer stable interactions compared to the control drug. This intermittent bonding pattern aligns with the lower electrostatic contribution seen in the binding free energy analysis, highlighting glabridin’s reliance on other non-covalent interactions for stability. The pinocembrin–protein complex exhibited the most consistent hydrogen bonding, forming 1–2 hydrogen bonds throughout the simulation. The stable and persistent hydrogen bonds indicated strong and reliable interactions, contributing to pinocembrin’s favorable binding affinity. The stability of these hydrogen bonds correlated with the intermediate binding free energy, reflecting the balance between electrostatic and van der Waals interactions.

#### 2.3.5. Dynamic Cross-Correlation (DCC) Analysis

The dynamic cross-correlation matrices (DCCM) obtained from molecular dynamics and simulations provide insights into the internal motions and correlated movements within the protein in different states. The DCCM plots are illustrated in Figure 5, with the color scale indicating the degree of correlation ranging from −1 (anti-correlated, blue) to +1 (correlated, red).

The apoprotein displayed significant regions of both positive and negative correlations. The extensive presence of red and blue regions across the matrix suggests highly dynamic interactions among various residues. The strong diagonal from the top left to the bottom right is indicative of the inherent structural integrity and intramolecular contacts. The broad distribution of correlated and anti-correlated motions in the apoprotein suggests that the protein in its apo form maintains a balance of flexible and rigid regions, which is typical for functional proteins needing to adopt multiple conformations. The control drug–protein complex exhibited an altered correlation pattern compared to the apo form. The overall intensity of correlations was reduced, which might indicate a stabilizing effect exerted by the control drug. The regions of high correlation were more localized (60–80 and 120–140), suggesting that the drug binding induces a more rigid structure, thereby reducing the dynamic flexibility of certain regions. This could imply that the control drug stabilizes the protein by constraining the movements of specific residues, potentially affecting the protein’s functional dynamics. In the glabridin–protein complex, there were noticeable changes in the dynamic correlations when compared to the apoprotein and the control drug complex. The pattern revealed distinct regions of altered correlations, implying that glabridin binding significantly influenced the protein’s dynamics. The presence of pronounced red and blue patches indicated that glabridin not only stabilized certain regions but also introduced new dynamic correlations and anti-correlations. This suggested that glabridin may induce conformational changes that could impact the protein’s functionality, enhancing or inhibiting its activity. The DCCM for the pinocembrin–protein complex showed a unique pattern of dynamic correlations. Compared to the apo and other complexes, pinocembrin induced a more heterogeneous distribution of correlated and anti-correlated motions. This could indicate that pinocembrin affected the protein in a more nuanced manner, modulating its flexibility and stability in specific regions. The distinct correlation patterns suggested that pinocembrin binding might promote specific functional conformations, potentially influencing the protein’s activity in a unique way compared to the control drug and glabridin.

#### 2.3.6. Principal Component Analysis and Free-Energy Calculations

To gain a deeper understanding of how ligand binding influences protein dynamics, six principal component analyses (PCA) were performed on the backbone coordinates of the apoprotein and protein–ligand complexes. The primary movements of the localized fluctuations were described by the first few principal components. Typically, the two most significant principal components (PC1 and PC2) represent over 70% of the total fluctuations for both the apoprotein and the protein–ligand complexes in our study. It was observed that the conformational space sampled by pinocembrin and glabridin complex systems was more restricted when compared to both the control drug and apoprotein. The free energy landscapes (FELs) were generated from the principal components (PC1 and PC2) and are shown in Figure 6. The plots represent the Gibbs free energy as a function of the two largest principal components.

The free energy landscape of the apoprotein showed a highly rugged surface with multiple local minima. The energy basins were distributed across a wide range of PC1 and PC2 values, indicating significant conformational diversity. The broad distribution of states suggests that the apoprotein explores a wide conformational space, reflecting its intrinsic flexibility in the absence of a ligand. The lowest energy state is centrally located, surrounded by higher energy barriers, which the protein may transition over during its dynamic motion. The control drug–protein complex exhibited a more localized free energy landscape compared to the apoprotein. The energy basins were concentrated in a narrower range of PC1 and PC2 values. This indicated that the binding of the control drug stabilized the protein into fewer conformational states. The reduction in conformational diversity suggested a restricted dynamic behavior, potentially due to the stabilizing interactions between the protein and the control drug. The primary energy minimum was slightly shifted compared to the apoprotein, indicating a conformational change upon ligand binding. The glabridin–protein complex displayed an intermediate level of conformational restriction. The free energy landscape showed a distinct pattern with energy basins that were more spread out than the control drug complex but less so than the apoprotein. The presence of glabridin leads to the stabilization of specific conformations, as evidenced by the defined energy basins, yet allows some degree of flexibility. This indicates that glabridin binding partially restricted the protein dynamics while maintaining a certain level of conformational exploration. The lowest energy state was significantly lower than those in the control and apo states, suggesting a stronger stabilizing effect by glabridin. The pinocembrin–protein complex showed a highly localized and distinct free energy landscape characterized by a narrow distribution of energy minima. This indicates a strong stabilizing effect of pinocembrin on the protein, significantly restricting its conformational flexibility. The lowest energy basin was well-defined and deeper compared to other complexes, suggesting that pinocembrin binding results in a highly stable protein conformation. The confinement of states within a limited region of PC1 and PC2 space implies that pinocembrin induced a more rigid structural state, reducing the protein’s dynamic behavior.

#### 2.3.7. Binding Free Energy Analysis

To understand the binding interactions and stability of the protein–ligand complexes, we performed a detailed binding free energy analysis. The results, including van der Waals (ΔVDWAALS), electrostatic energy (ΔEEL), polar solvation energy (ΔEPB), non-polar solvation energy (ΔENPOLAR), gas–phase interaction energy (ΔGGAS), solvation free energy (ΔGSOLV), and the overall binding free energy, are summarized in Table 3. The binding free energy analysis showed distinct differences in affinities for the control drug, glabridin, and pinocembrin complexes. The control drug had the weakest binding affinity (−10.23 kcal/mol) due to high polar solvation energy despite strong electrostatic interactions. Glabridin exhibited the strongest binding affinity (−20.83 kcal/mol) with favorable van der Waals and electrostatic interactions and relatively low polar solvation energy. Pinocembrin had an intermediate binding affinity (−17.06 kcal/mol) with favorable van der Waals interactions and moderate electrostatic contributions.

### 2.4. Physicochemical Properties Analysis

To evaluate the drug-likeness and bioavailability of the compounds, we analyzed various physicochemical properties, including molecular weight, polar surface area, number of hydrogen bond donors and acceptors, MLOGP (logarithm of partition coefficient between n-octanol and water), Lipinski’s rule of five, Veber’s rule, bioavailability score, PAINS (pan assay interference compounds) alerts, and lead-likeness. The data are summarized in Table 4. Both pinocembrin and glabridin complied with Lipinski’s rule of five and Veber’s rule, indicating favorable drug-likeness. However, pinocembrin and the control drug exhibited PAINS (pan assay interference compounds) and lead-likeness, while glabridin did not show PAINS.

## 3. Discussion

Phytocompounds from plants exhibit diverse anti-bacterial properties, making them valuable in combating microbial infections. Carvacrol and thymol, found in oregano and thyme oils, respectively, demonstrate strong activity against both Gram-positive and Gram-negative bacteria [27]. Cinnamaldehyde from cinnamon bark shows efficacy against multidrug-resistant strains, while eugenol in clove oil targets oral pathogens [28,29]. Berberine, derived from plants like Berberis species, has broad-spectrum anti-bacterial effects useful for gastrointestinal infections [30]. Curcumin, from turmeric, exhibits anti-bacterial properties against various pathogens [31]. Glabridin in licorice root and epigallocatechin gallate (EGCG) in green tea also display anti-bacterial activity especially, particularly against oral bacteria [32,33,34]. These compounds offer promising alternatives or complements to conventional antibiotics, highlighting their potential in therapeutic applications against bacterial infections.

Our study investigated the interactions between the SrtA protein of *E. faecalis* and phyto-compounds with anti-bacterial activity. The Tanimoto similarity heatmap demonstrated the structural relationships among thirty natural phyto-compounds and ampicillin, highlighting clusters of high similarity, such as Asiatic acid, betulinic acid derivative, oleanolic acid, and ursolic acid, as well as the hydroxyflavanones, indicating shared structural motifs. Unique compounds like gamma-terpinene and curcumin showed lower similarity, suggesting diverse biological activities. These findings imply that structurally analogous compounds may possess comparable inhibitory potential against the SrtA protein of *E. faecalis*.

The binding energy and interaction analysis by molecular docking of various phytocompounds with SrtA protein revealed significant insights into their potential as anti-bacterial agents. Pinocembrin exhibited the highest binding affinity, suggesting strong interactions with the target protein, reinforced by key hydrophobic interactions and hydrogen bonds. Compounds such as glabridin and ursolic acid also demonstrated notable binding energies, with multiple hydrophobic interactions and hydrogen bonds enhancing their binding stability. Curcumin, eriodictyol, and 7-hydroxyflavanone also showed robust binding energies and interaction profiles, indicating their potential effectiveness. In contrast, ampicillin, a commonly used antibiotic, showed a lower binding energy of −6.5 kcal/mol, suggesting weaker interactions compared to these phytocompounds. The comprehensive interaction profile of pinocembrin and glabridin underscores their potential as potent anti-bacterial agents, warranting further investigation and development as a novel anti-bacterial compound. 

Molecular dynamics simulation analysis revealed that pinocembrin induced the least conformational changes, while glabridin exhibited intermediate effects compared to the control drug. Pinocembrin stabilized the N-terminal region effectively, as evidenced by reduced fluctuations. Free energy landscape analysis indicated the stabilizing effects of pinocembrin on the protein structure, with a highly localized energy basin. Hydrogen bond analysis supported favorable interactions between pinocembrin and the protein. The physicochemical properties analysis highlighted the drug-likeness of pinocembrin and glabridin. In summary, our findings provide valuable insights into the molecular interactions and dynamic behavior of SrtA protein with selected compounds, laying the groundwork for potential drug discovery efforts against *E. faecalis* infections.

Two previous studies employed computational methods to identify inhibitors for *E. faecalis* SrtA. Selvaraj et al. employed a virtual screening approach to sift through compounds from DrugBank [35]. However, their focus primarily centered on broad-spectrum antibiotics, which are known to contribute to resistance development. Another study by Sivaramakrishnan et al. similarly utilized a virtual screening approach but focused exclusively on curcumin analogs targeting the *E. faecalis* SrtA protein. This study did not include extended molecular dynamics and simulation analyses to assess the stability of the curcumin analogs [36]. In contrast, the current study explored diverse classes of phytocompounds known for their effectiveness against *E. faecalis*. Furthermore, extended molecular dynamics and simulation analyses were employed to validate the stable conformations of the highly potent phytocompounds.

The present study highlights promising alternatives or complements to conventional antibiotics, emphasizing their potential in therapeutic applications against bacterial infections. These findings require further in vitro validation to confirm their efficacy and safety.

## 4. Materials and Methods

### 4.1. Protein Structure Modeling and Validation

The SrtA protein sequence of *E. faecalis* (Accession ID: WP_010715428) was obtained from NCBI [37]. The protein structure was modeled using MODELLER 10.4 [38]. A suitable template for SrtA structure prediction was identified using BLASTP [39]. The SrtA of *B. anthracis* (PDB ID–2KW8) was chosen as a template for predicting *E. faecalis* structure based on the sequence identity and E-value [40]. Ten models were predicted and evaluated using the molecular probability density function (molpdf) and Discrete Optimized Protein Energy (DOPE) score. Additionally, loop refinement was performed with MODELLER 10.4. The final SrtA protein structure was visualized in PyMOL 2.5.5 (The PyMOL Molecular Graphics System, Version 3.0 Schrödinger, LLC New York, NY, USA) and further validated using the ERRAT [41], PROCHECK [42], and ProSA-web servers [43]. The reliability of the modeled protein was assessed by superimposing it against the template protein in PyMOL and calculating the RMSD. The 3D coordinates of the modeled proteins were analyzed for dihedral angle distribution using a Ramachandran plot in PROCHECK and nonbonded atomic interactions using the ERRAT tool of the SAVES server. Furthermore, the PROSA web server evaluated the quality of the model based on Cα positions.

### 4.2. Protein Preparation

The modeled SrtA protein was prepared for molecular docking using AutoDockTools-1.5.7 [44]. The protein structure processing involved the addition of polar hydrogens, non-polar hydrogen merging, and Kollman charges. In addition, the presence of missing atoms was also checked. 

### 4.3. Ligand Identification and Selection

Natural products are a rich source of SrtA inhibitors. After an initial literature review, around thirty phytocompounds were selected based on their proven antibiofilm activity against *E. faecalis*. The details of compounds used in the current study are listed in Table 5. Ampicillin was used as the control drug in our docking study targeting the SrtA protein of *E. faecalis*. Ampicillin’s clinical efficacy in treating endodontic infections caused by *E. faecalis* has been well-documented in various studies, highlighting its role as a reliable therapeutic option in endodontic practice [45,46]. This provided a reliable benchmark for assessing the binding affinities of the investigated phytocompounds.

The structures of these compounds were obtained from PubChem [47] and geometry-optimized using the Discovery Studio 2021 Client version. The ligands were then processed by adding polar hydrogens and incorporating Gasteiger charges using the Autodock Raccoon tool [48]. 

**Table 5 ijms-25-07727-t005:** Structure, class, and antibiofilm activity against *E. faecalis*.

Sl. No	Compound	Structure	Class	Activity	References
1	Gamma-Terpinene	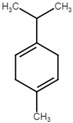	Hydrocarbon monoterpene	BEC50 = 418 μg/mL	[29]
2	p-cymene	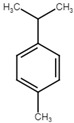	Hydrocarbon monoterpene	BEC50 = 636 μg/mL	[29]
3	Carvacrol	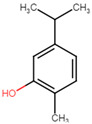	Phenolic monoterpene	BEC50 = 309 μg/mL	[29,49]
4	Thymol	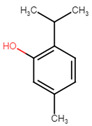	Phenolic monoterpene	BEC50 = 186 μg/mL	[29]
5	Asiatic acid	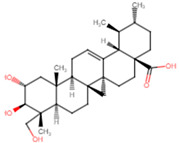	Pentacyclic triterpene	Reduction of biofilm formation and bacterial cell counts at 0.75 × MIC	[50]
6	Betulinic acid derivative—succinyl ester	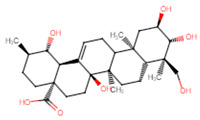	Pentacyclic triterpene	Reduction of biofilm formation at 25 and 100 μM of BA-succinyl ester	[51]
7	Oleanolic acid	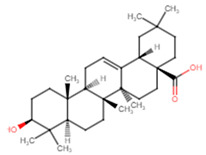	Pentacyclic triterpene	Reduction of biofilm formation by >50% at 62.5 μg/mL	[52]
8	Ursolic acid	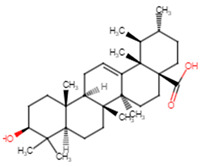	Pentacyclic triterpene	Inhibition of biofilm formation at 100 μM	[51]
9	Cinnamaldehyde	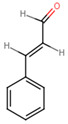	Polyphenol	Diminished biofilm viability (72-h old) at 0.5, 0.75 and 1% wt/vol	[53]
10	Eugenol	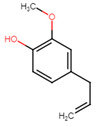	Polyphenol	BEC50 = 734 μg/mL	[29]
11	Epigallocatechin gallate	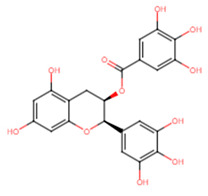	Flavanol	Significant reduction of biofilm cell counts at 125 and 250 μg/mL	[54]
12	Proanthocyanidins	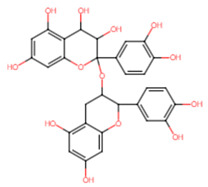	Flavanol	>50% reduction of biofilm formation at 62.5 μg/mL	[55]
13	Quercetin	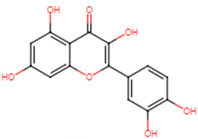	Flavanol	70–95% reduction in biofilm formation at 0.125–0.5 × MIC (=64–256 μg/mL)	[56]
14	Myricitrin	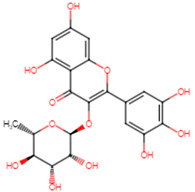	Flavonol	>60% reduction in biofilm formation at 250 μg/mL	[57]
15	Glabridin	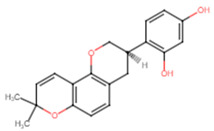	Isoflavonoid	Reduction in biofilm metabolic activity by 11% at 25 μg/mL	[58]
16	Licoricidin	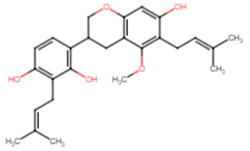	Isoflavonoid	Reduction in biofilm formation by 31.6% at 0.25 × MIC = 1.56 μg/mL	[59]
17	Licochalcone A	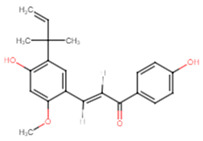	Chalcone	Reduction in biofilm metabolic activity by 29% at 12.5 μg/mL	[58]
18	Curcumin	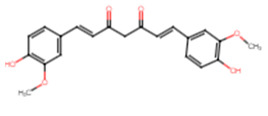	Curcuminoid	Biofilm disruption by 31.6%	[60]
19	Berberine	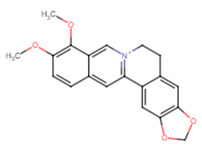	Alkaloid	Reduction of biofilm formation at (60, 80 and 100 μg/mL)	[61]
20	2′-hydroxyflavanone	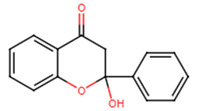	Flavanone	Not determined	[23]
21	3′-hydroxyflavanone	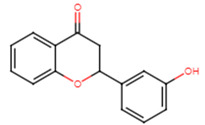	Flavanone	MIC = 256 μg/mL	[23]
22	4′-hydroxyflavanone	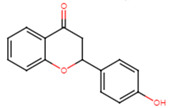	Flavanone	Not determined	[23]
23	6′-hydroxyflavanone	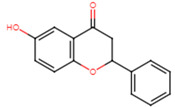	Flavanone	Not determined	[23]
24	7′-hydroxyflavanone	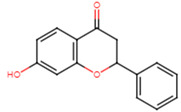	Flavanone	Not determined	[23]
25	Pinocembrin	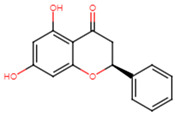	Flavanone	MIC > 512 μg/mL	[23]
26	6,2′-dihydroxyflavanone	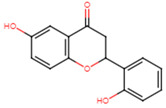	Flavanone	MIC = 256 μg/mL	[23]
27	Naringenin	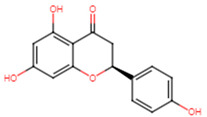	Flavanone	MIC = 256 μg/mL	[23]
28	Eriodictyol	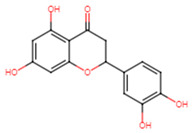	Flavanone	MIC = 256 μg/mL	[23]
29	Dihydrorobinetin	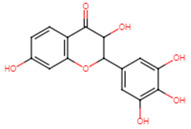	Flavanone	MIC = 256 μg/mL	[23]
30	Taxifolin	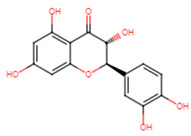	Flavanone	MIC = 128 μg/mL	[23]

### 4.4. Molecular Docking

The molecular docking analysis was performed using Autodock Vina [62]. The active site of the SrtA was predicted using the Computed Atlas of Surface Topography of proteins (CASTp) web server [63]. The active site residues includes VAL88, ILE89, LEU91, GLU93, GLU95, ARG96, SER98, VAL99, ILE102, GLN103, SER104, PHE109, GLU111, SER113, VAL116, GLU121, ASP123, ILE125, GLN126, LEU127, PRO130, ASP135, SER136, PHE137, LEU141, TYR142, VAL147, LEU148, LYS150,THR 152, LYS155, ASN164, ALA165, TYR167, GLU168, GLY169, LEU170, LEU171, VAL178, SER179, VAL180, LEU183, LYS185, GLU200, GLN201, ASP205, MET206 THR210, MET211, LEU212, ASN213, LEU214, THR215, THR219, ALA228, THR229, LYS230, THR232, ILE238, ALA239, GLU240. The grid box was generated over the predicted binding site of the protein. For each ligand, 10 conformational poses were generated. These ligands were then ranked based on the binding energy of the highest-ranked conformation. Protein–ligand interactions were analyzed with Discovery Studio 2021 Client version, and the complex structures were visualized using PyMOL 2.5.5. Ampicillin was used as positive control to validate the docking protocol. The top two compounds with the high binding affinity, pinocembrin and glabridin, were subjected to molecular dynamics and simulation analysis.

### 4.5. Molecular Dynamics and Simulation

Molecular dynamics simulations (MDS) of apo–SrtA, SrtA–ampicillin, and SrtA–phytocompound complexes were performed to assess the stability of these systems at the atomic level. The MDS was conducted using the CHARMM27 force field in GROMACS 2024.2 [64], with ligand topology files generated via SwissParam [65]. Each system was solvated in the simple point charge (SPC) water model and placed within a dodecahedron box of 7.05774 nm³ volume, maintaining a 1.0 nm buffer from the box edges. Sodium ions were added to neutralize the systems, which were then energy minimized using the steepest descent algorithm for 50,000 steps, reaching an energy cutoff of 10.0 kJ/mol. Equilibration was conducted under both constant number of particles, volume, and temperature (NVT) and constant number of particles, pressure, and temperature (NPT) conditions for 1 ns at 300 K. Pressure and temperature were controlled using the Parrinello–Rahman barostat and the Berendsen thermostat, respectively (Berendsen et al., 1984; Parrinello and Rahman, 1981). Covalent bonds were constrained using the Linear Constraint Solver (LINCS) method. The production MDS was executed for 100 ns, employing the particle mesh Ewald (PME) method to manage long range electrostatic interactions. During the final MD run, coordinates were recorded at 10 ps intervals. The resulting trajectories were analyzed to calculate root mean square deviation (RMSD), root mean square fluctuation (RMSF), radius of gyration (Rg), solvent-accessible surface area (SASA), and hydrogen bond counts for the protein–ligand complexes.

#### 4.5.1. Essential Dynamics and Gibbs Energy Calculation

The trajectory files obtained from MD simulations were utilized to investigate the dominant motions in apoprotein and protein–ligand complexes through principal component analysis (PCA) or essential dynamics. We calculated the positional covariance matrix of atomic coordinates and its eigenvectors. The matrix was diagonalized using an orthogonal coordinate transformation matrix, resulting in a diagonal matrix of eigenvalues. The first eigenvector and its corresponding eigenvalue typically indicate the principal component of the trajectory, representing the main global motion of the structures. Free energy landscapes were obtained by plotting the first two principal components (PC1 and PC2) against each other, derived from the PCA of the trajectories for each system. The corresponding Gibbs energy reflects the conformations of molecules throughout the trajectory. Deep valleys represent stable and dominant conformations, while boundaries indicate intermediate conformations of the molecules. The gmx sham function in GROMACS was used for Gibbs energy calculations. The dynamic cross-correlation matrix (DCCM) was constructed to identify correlated motions of residues. The matrix (Cij) shows the time-correlated information between the i and j atoms of a protein. To construct the matrix, only Cα atoms from the last 500 snapshots were selected at 0.002 ns time intervals. Positive values indicate motions in the same direction (correlated motions), whereas negative values indicate atomic displacement in opposite directions. The DCCM plots were generated using MD-TASK [66].

#### 4.5.2. MM–PBSA Calculation

The binding free energy of the protein–ligand complex was calculated using the MM/PBSA method on the last 20 ns of the molecular dynamics (MD) simulation trajectory, conducted with GROMACS and processed using the gmx_MMPBSA plugin [67]. The system was prepared and equilibrated before the production run, and two hundred snapshots were extracted at 100 ps intervals from the final 20 ns. The MM/GBSA method combined molecular mechanics (MM) energy calculations, the Generalized Born (GB) model for polar solvation energy, and surface area-dependent terms for non-polar solvation energy. The gas–phase interaction energy between the protein and ligand was calculated using CHARMM forcefield. The Generalized Born (GB) model was used to estimate the polar solvation energy. The non-polar solvation energy was estimated using a surface area-dependent term. The binding free energy (ΔGbind) was calculated using the equation
ΔGbind = ΔEMM + ΔGSolv − TΔS
where ΔEMM is the molecular mechanics energy, ΔGSolv is the solvation free energy, and TΔS is the entropy term. The binding free energy values from the individual frames were averaged to obtain the final estimate. The standard deviation and error estimates were calculated to evaluate the reliability and convergence of the results.

### 4.6. Assessment of Physicochemical, Drug-Likeness, and ADMET Properties of Phytocompounds

The functional activity-determining physicochemical properties of the top two phytocompounds were evaluated using the SwissADME server [68]. This included properties such as molecular weight (MW), total polar surface area (TPSA), number of hydrogen bond donors (HBD), number of hydrogen bond acceptors (HBA), and octanol/water partition coefficient (LogP). The evaluation also assessed drug-likeness and oral availability in accordance with Lipinski’s rule of five and Veber’s rule, respectively.

## 5. Conclusions

In conclusion, our comprehensive study delved into the intricate interactions between the SrtA protein of *E. faecalis* and various phytocompounds, aiming to shed light on potential avenues for drug discovery against *E. faecalis* infections. Through molecular docking analysis, we identified promising binding affinities for several compounds, with pinocembrin emerging as particularly noteworthy due to its strong interactions. Our subsequent analyses, including molecular dynamics simulations, free energy landscape analysis, and hydrogen bond analysis, provided deeper insights into the dynamic behavior and stability of the protein–ligand complexes. Pinocembrin exhibited a remarkable ability to stabilize the protein structure, induce minimal conformational changes, and form stable hydrogen bonds, indicating its potential as a lead compound for further drug development efforts. Furthermore, our physicochemical properties analysis highlighted the favorable drug-likeness of pinocembrin and glabridin, underscoring their potential as viable candidates for future therapeutic interventions. Overall, our findings pave the way for continued exploration and optimization of these compounds as potential anti-*E. faecalis* agents, offering new hope in the fight against bacterial infections.

## Figures and Tables

**Figure 1 ijms-25-07727-f001:**
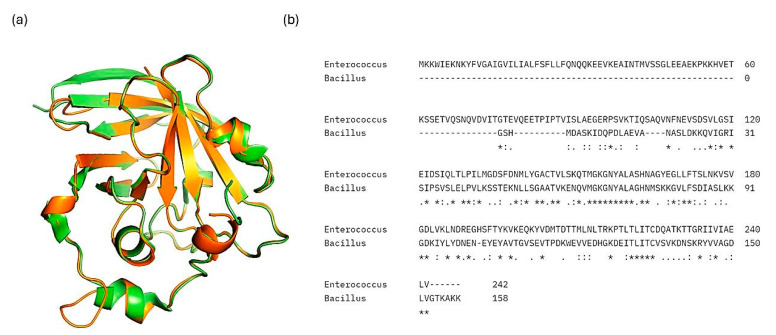
(**a**) The superimposition of template (green) and modeled SrtA protein (orange). (**b**) Pairwise sequence alignment of the SrtA protein sequences of *E. faecalis* and *B. anthracis*. (. *) indicate identical amino acid, (:) indicate conserved substitution, (.) indicate semi-conserved substitution and (-) indicate Gap.

**Figure 2 ijms-25-07727-f002:**
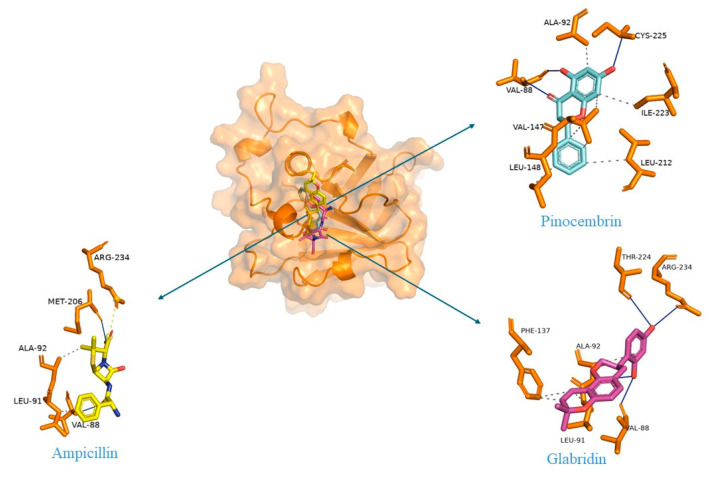
Docking analysis and visualization of control drug (ampicillin), pinocembrin, and glabridin complexes, respectively. Dashed lines indicate hydrophobic interactions, and solid lines indicate hydrogen bonding. Yellow dashed lines indicate salt bridges.

**Figure 3 ijms-25-07727-f003:**
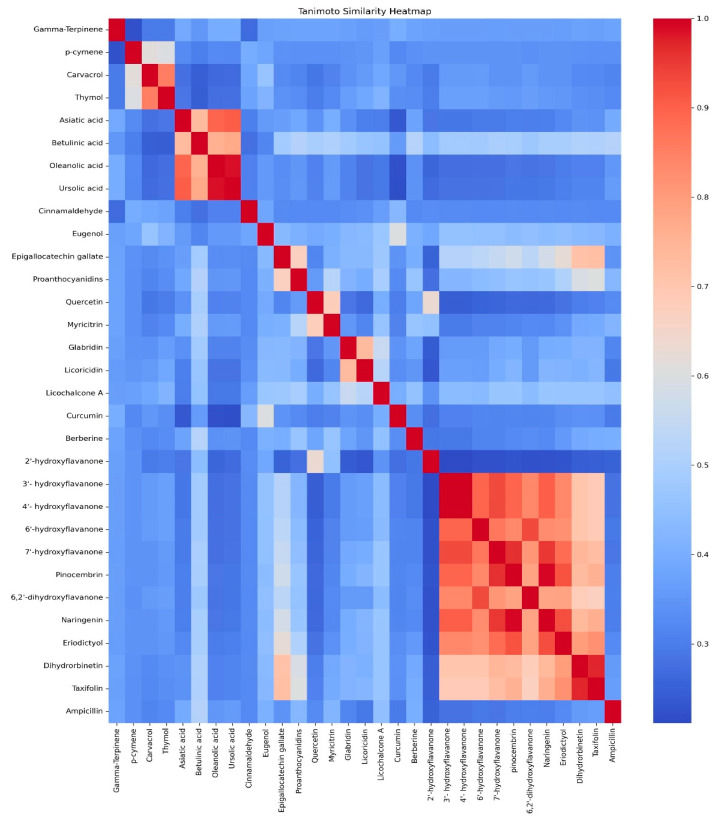
Heatmap of Tanimoto similarity index of the ligands from our study.

**Figure 4 ijms-25-07727-f004:**
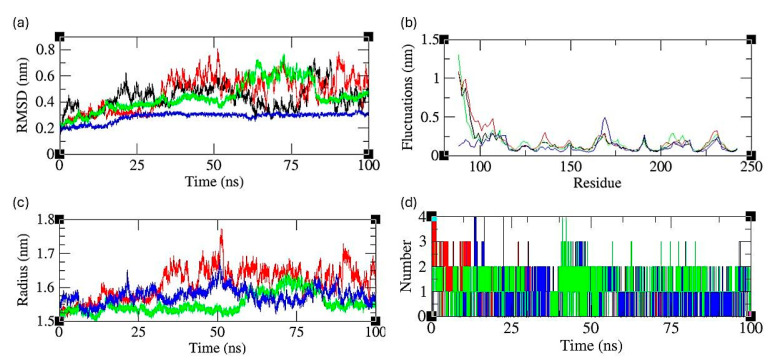
Analysis of (**a**) RMSD plot, (**b**) RMSF, (**c**) Radius of Gyration, and (**d**) Hydrogen bonds. Black curve—apoprotein, red curve—control drug complex, blue curve—glabridin complex, and green curve—pinocembrin complex.

**Figure 5 ijms-25-07727-f005:**
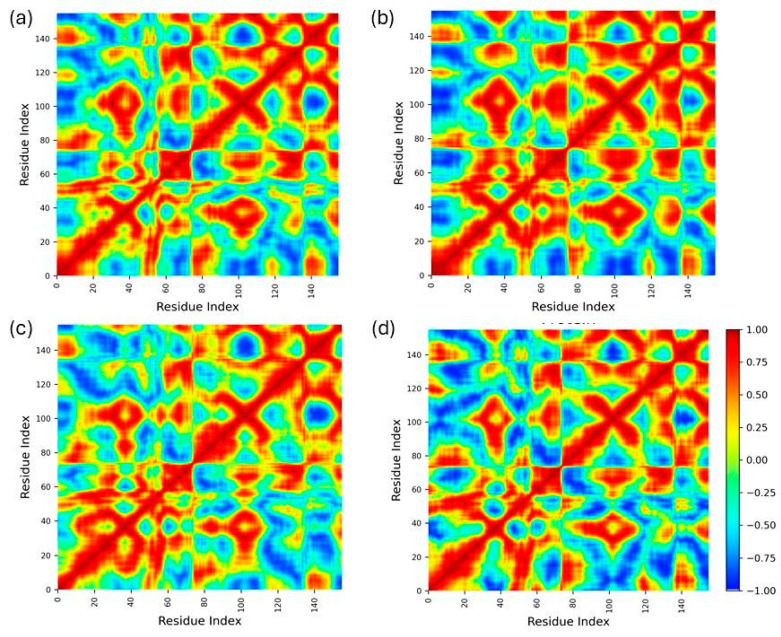
Dynamic cross-correlation map (DCCM). The DCCM map for (**a**) apoprotein, (**b**) control drug, (**c**) glabridin, and (**d**) pinocembrin complexes, respectively. The map shows the correlated motions of protein residues in apo and complex systems. The red color represents a positive correlation, and the blue color represents a negative correlation. The color gradients represent a gradual decrease in the correlation.

**Figure 6 ijms-25-07727-f006:**
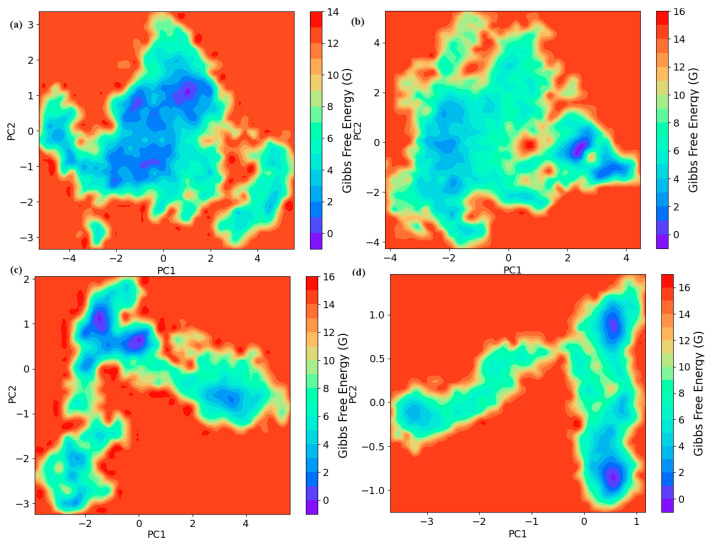
Free energy landscape values of apoprotein and protein–ligand complexes. (**a**) apoprotein, (**b**) control drug, (**c**) glabridin, and (**d**) pinocembrin complexes, respectively. The color bar represents the relative free energy value in kcal/mol.

**Table 1 ijms-25-07727-t001:** Structural validation of top three SrtA models generated by Modeller.

Model	ERRAT	Ramachandran Plot (%)	ProSA-Z Score	DOPE	molpdf
Model 1	64.287	98.6	−6.1	1303.28	−17,533.61
Model 2	47.619	98.6	−3.9	1248.24	−17,238.74
Model 3	42.6901	98.4	−4.5	1261.07	−17,197.06

**Table 2 ijms-25-07727-t002:** Binding energy and amino acid interaction table between the phyto-compounds and SrtA protein.

Si. No	Phytocompounds	Binding Energy (kcal/mol)	Hydrophobic Interactions	Hydrogen Bonds	pi-Cation Interactions	Salt Bridge
1	Pinocembrin	−8.0	ALA92, VAL147, LEU148, LEU212, ILE223	VAL88, CYS225		
2	Glabridin	−7.9	LEU91, ALA92, PHE137	VAL88, ALA92, THR224, ARG234		
3	Ursolic acid	−7.9	ALA92, VAL147, LEU148, THR209, LEU212, ILE223			
4	7-hydroxyflavanone	−7.9	GLN227, LYS230, ARG234	GLU95		
5	Epigallocatechin3-gallate	−7.8	VQL147, LEU148, THR209, LEU212	ASP135, LUE212, ASN213, ARG234		
6	Curcumin	−7.7	LEU91, ASP135, MET206, GLN227	ASP135, GLN227, ARG234		
7	Eriodictyol	−7.7	MET206, LYS230	ILE89, GLU93, GLU95, CYS225, THR232		
8	Myricitrin	−7.7	VAL147, THR209	LEU91, MET206, THR224, ARG234		
9	6-hydroxyflavanone	−7.7	GLN227, LYS230, ARG234			
10	Naringenin	−7.5	GLU93, MET206, LYS230	ILE89, GLU95, GLN227		
11	6,2′-dihydroxyflavanone	−7.5	GLN227, LYS230, ARG234			
12	Asiatic acid	−7.5	ALA92, VAL147, LEU148, THR209, LEU212	ARG234		
13	Oleanolic acid	−7.4	ILE89, ALA92, VAL147, LEU212, ILE223			
14	Quercetin	−7.3	LEU91, ALA92	ASP135, MET206, ARG234		
15	2-hydroxyflavanone	−7.2	ILE89, LYS230, ARG234	GLU95	LYS230	
16	Dihydrorobinetin	−7.2	GLU93, LYS230, LYS230	GLU95, GLN227		
17	Licochalcone A	−7.2	LUE91, ALA92, VAL147, GLN227	VAL88, ASP226, GLN227		
18	Licoricidin	−7.2	ILE89, LEU91, ALA92, THR209, ILE223	ARG234		
19	Proanthocyanidins	−7.2	LUE91, ASP135, VAL147, LEU212	ILE89, MET133, ASP135, SER149, ASN213		
20	Betulinic acid derivative	−7.2	VAL88, ALA92, LEU212	VAL88, ARG234		
21	Berberine	−7.1	LEU91, PHE137			
22	3-hydroxyflavanone	−6.8	LYS230	GLN227		
23	Taxifolin	−6.8	LEU91	ASP135		
24	4-hydroxyflavanone	−6.8	LYS230	GLN227		
25	Ampicillin	−6.5	LEU91, ALA92	VAL88, MET206		ARG234
26	Carvacrol	−5.5	ALA92, MET206, LYS230	GLN227, THR229		
27	Thymol	−5.5	LEU148, GLN151, LYS155, LEU214	LEU212		
28	Eugenol	−5.2	LEU148, LEU214	LYS155		
29	Cinnamaldehyde	−5.0	LEU148, LEU214	LYS155		
30	P-cymene	−5.0	LEU148, GLN151, LYS155, LEU214			
31	Gamma terpinene	−4.9	LEU148, GLN151, LYS155, LEU214			

**Table 3 ijms-25-07727-t003:** MMPBSA binding free energy of the protein–ligand complexes.

Complexes	ΔVDWAALS	ΔEEL	ΔEPB	ΔENPOLAR	ΔGGAS	ΔGSOLV	Binding Free Energy *
Control drug	−30.03	−182.43	205.23	−3	−212.46	202.23	−10.23
Glabridin	−35.6	−21.31	39.67	−3.6	−56.91	36.07	−20.83
Pinocembrin	−30.81	−12.66	29.38	−2.97	−43.48	26.41	−17.06

ΔVDWAALS—van der Waals energy, ΔEEL—electrostatic energy, ΔEPB—polar solvation energy in Poisson–Boltzmann method, ΔENPOLAR—non-polar solvation energy in Poisson–Boltzmann method, ΔGGAS—gas–phase molecular mechanics free energy, ΔGSOLV—solvation free energy. * Kcal/mol.

**Table 4 ijms-25-07727-t004:** Physicochemical properties of the top hit compounds and control drug.

Compounds	MW	PSA	HBD	HBA	MLOGP	Rule 1	Rule 2	Bioavailability Score	PAINS	Score
Pinocembrin	256.25	66.76	2	4	1.27	Yes	Yes	0.55	0	Yes
Glabridin	324.37	58.92	2	4	2.73	Yes	Yes	0.55	0	No
Ampicillin	349.4	138.03	3	5	0.75	Yes	Yes	0.55	0	Yes

MW—molecular weight, PSA—polar surface area, HBD—hydrogen bond donors, HBA—hydrogen bond acceptors, MLOGP—logarithm of partition coefficient between n-octanol and water, Rule 1—Lipinski’s rule of five, Rule 2—Veber’s rule, PAINS—pan assay interference compounds, and score—lead-likeness.

## Data Availability

All data are available online. No unpublished data have been used in this paper.

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
