# Peer review of "Exploring Plant-Based Compounds as Alternatives for Targeting Enterococcus faecalis in Endodontic Therapy: A Molecular Docking Approach"

_ijms, 2024, doi:10.3390/ijms25147727_

Round 1

Reviewer 1 Report

Comments and Suggestions for Authors

The manuscript IJMS 3073202, authors reported the plant-based compounds antibacterial activity specifically against Enterococcus faecalis by molecular docking. Authors have to revise the manucript carefully and provide some stong evidences from literature whih can support their proposed molecular docking study.

1. The authors have to prove that what was the chemical nature of each compound? 

2. It would be better if the author provide some preliminary in-vitro experimental data about pinocembrin or other compounds, which would strengthen the molecular docking proposal. 

3.  This study proposed new plant-based compound for bacterial treatment. though it required further in-vitro experimentation. As it is an AI-based study, it needs experimental confirmation. 

4. Authors need to support their data by experiments.  

5. Also improve the discussion part keeping in mind to support your molecular docking resutls.

6. Conclusions are in line with the molecular-based study. 

7. References are accurate.

Comments on the Quality of English Language

Moderate to minor revision are required 

Author Response

Exploring Plant-Based Compounds as Alternatives for Targeting Enterococcus faecalis in Endodontic Therapy: A Molecular Docking Approach

Manuscript ID: ijms-3073202

We thank all the reviewers and acknowledge the comments made by all the reviewers.

Answers to reviewer 1 comments

Query 1: The authors have to prove that what was the chemical nature of each compound?

Answer 1: Thank you and we acknowledge the comment. As per the suggestion given by the reviewer, an additional table , Table 5 lists the chemical nature, structure and references is included in the revised manuscript.

Query 2: It would be better if the author provide some preliminary in-vitro experimental data about pinocembrin or other compounds, which would strengthen the molecular docking proposal.

Answer 2: Thank you and we acknowledge the comment. The details about the activity of the Phytocompounds are listed in Table 5.

Query 3: This study proposed new plant-based compound for bacterial treatment. though it required further in-vitro experimentation. As it is an AI-based study, it needs experimental confirmation. 

Answer 3: Thank you for your valuable feedback. We acknowledge that while our study proposes new plant-based compounds for bacterial treatment through AI-based computational analysis, further in-vitro experimentation is essential to validate these findings. We plan to conduct comprehensive in-vitro assays to test the efficacy and safety of the identified compounds against the target bacteria.

We have included “These findings require further in-vitro validation to confirm their efficacy and safety.” in the current manuscript.

Query 4: Authors need to support their data by experiments.  

Answer 4: Thank you for your comment. The work is planned as suggested by the reviewer.

Query 5: Also improve the discussion part keeping in mind to support your molecular docking results.  

Answer 5: Thank you and we acknowledge your comment. The discussion part is revised as per the suggestions of the reviewers.

Reviewer 2 Report

Comments and Suggestions for Authors

The present study, "Exploring Plant-Based Compounds as Alternatives for Targeting Enterococcus faecalis in Endodontic Therapy: A Molecular Docking Approach ," performs a in-silico study to determine the anti-E. faecalis agents (Pinocembrin and Glabridin) targeting SrtA. Overall, the work is very significant in view of the fight against bacterial infections, but several concerns in the manuscript need to be addressed for further processing of this manuscript.

Comments
1.Please write the key properties off the selected 30 natural phyto-compounds in this study. Moreover this study didnot perform only binding affinity and stability not inhibitory potential. Please correct

2. The target protein WP_01071542 is 119 amino acid in NCBI but Figure 1 indicates the target protein 242 amino acid. Please explain.

3.  Why only 30 natural phyto-compounds against the SrtA protein?
Authors are also suggested to provide a table with these compounds and their inhibitory concentrations.

4. How did you select ampicillin as a positive control. Is there any report for the activity of ampicillin against SrtA protein. What are the MIC of  Ampicillin against the Enterococcus faecali.

5. Please remove the repetition " The active site of the SrtA was predicted using Computed Atlas of Surface Topography of proteins (CASTp) web server [31]. The grid box was generated over the predicted binding site of the protein"

6. Why did two ligands bind in two different places on the protein? Does it have two active site? Please recheck the  castp analysis active site residues (residues 88-240) and verify with other tools including the literature reports.

7. Please discuss the results presented in Figure 3.

8. Line 449 Mention the top two compounds with the high binding affinity

9. Line 539, 549 : Remove recheck the typos. There are several language errors in the manuscript. Please correct

10. There is no discussion in this manuscript. Please discuss your finding and compares with the previous researches. Please point out the significant findings of this study.

Comments on the Quality of English Language

 Minor editing of English language required

Author Response

Exploring Plant-Based Compounds as Alternatives for Targeting Enterococcus faecalis in Endodontic Therapy: A Molecular Docking Approach

Manuscript ID: ijms-3073202

We thank all the reviewers and acknowledge the comments made by all the reviewers.

Answers to reviewer 2 comments

Query 1: Please write the key properties off the selected 30 natural phyto-compounds in this study. Moreover this study didnot perform only binding affinity and stability not inhibitory potential. Please correct

Answer 1: Thank you and we acknowledge the comment. We corrected the sentence as per your suggestion.

Query 2: The target protein WP_01071542 is 119 amino acid in NCBI but Figure 1 indicates the target protein 242 amino acid. Please explain.

Answer 2: The protein ID is WP_010715428, as mentioned in the manuscript. Kindly use the link to verify the details of the SrtA  protein. https://www.ncbi.nlm.nih.gov/protein/WP_010715428

Query 3.a: Why only 30 natural phyto-compounds against the SrtA protein?

Answer 3.a: Thank you and we acknowledge the comment. After thorough literature search we identified the 30 phytocompounds which have proven activity against E. faecalis, hence we used the compounds for our analysis.

Query 3.b: Authors are also suggested to provide a table with these compounds and their inhibitory concentrations.

Answer 3.b: Thank you and we acknowledge the comment. As per the reviwer’s suggestion, we have included Table 5 with the details of the phytocompounds in our revised manuscript.

Query 4. How did you select ampicillin as a positive control. Is there any report for the activity of ampicillin against SrtA protein. What are the MIC of  Ampicillin against the Enterococcus faecali.

Answer 4: Thank you and we acknowledge the comment. We observed that majority of the E. faecalis strains were resistant to major antibiotics, many studies have documented their susceptibility to ampicillin, hence we have used ampicillin as our control drug. In one such study by by Metzidie et al., the MIC of ampicillin were 3, 54, 91 and 100  at 0.25, 0.5, 1 and 2 mg/L.

Reference:

  1. Evgenia Metzidie, Evangelos N. Manolis, Spyros Pournaras, Danai Sofianou, Athanassios Tsakris, Spread of an unusual penicillin- and imipenem-resistant but ampicillin-susceptible phenotype among Enterococcus faecalis clinical isolates, Journal of Antimicrobial Chemotherapy, Volume 57, Issue 1, January 2006, Pages 158–160, doi.org/10.1093/jac/dki427.
  2. Conceição N, de Oliveira Cda C, da Silva LE, de Souza LR, de Oliveira AG. Ampicillin susceptibility can predict in vitro susceptibility of penicillin-resistant, ampicillin-susceptible Enterococcus faecalis Isolates to amoxicillin but not to imipenem and piperacillin. Journal of Clinical Microbiology. Volume 50, Issue 11, November 2012, Pages729-731, doi: 10.1128/JCM.01246-12.

Query 5: Please remove the repetition " The active site of the SrtA was predicted using Computed Atlas of Surface Topography of proteins (CASTp) web server [31]. The grid box was generated over the predicted binding site of the protein"

Answer 5: Thank you and we acknowledge the comment. The repeating sentences are removed.

Query 6: Why did two ligands bind in two different places on the protein? Does it have two active site? Please recheck the  castp analysis active site residues (residues 88-240) and verify with other tools including the literature reports.

Answer 6: Thank you and we acknowledge the comment. The castp active site residues are verified and included in the Materials and methods section under molecular docking section of the manuscript. And Figure 2 is also corrected in the revised manuscript.

Query 7: Please discuss the results presented in Figure 3.

Answer 7: Thank you and we acknowledge the comment. As suggested we have added the discussion regarding Figure 3 in the revised manuscript.

Query 8: Line 449 Mention the top two compounds with the high binding affinity.

Answer 8: Thank you and we acknowledge the comment. As suggested we have added “The top two compounds Pinocembrin and Glabridin with the high binding affinity were subjected to molecular dynamics and simulation analysis.” in the revised manuscript.

Query 9: Line 539, 549 : Remove recheck the typos. There are several language errors in the manuscript. Please correct

Answer 9: Thank you and we acknowledge the comment. As suggested we have corrected all the typos and language error in the revised manuscript.

Query 10: There is no discussion in this manuscript. Please discuss your finding and compares with the previous researches. Please point out the significant findings of this study.

Answer 10: Thank you and we acknowledge the comment. As suggested we have compared previous studies with our current study and included in the discussion section.

Reviewer 3 Report

Comments and Suggestions for Authors

The paper entitled 'Exploring Plant-Based Compounds as Alternatives for Target-2 ing Enterococcus faecalis in Endodontic Therapy: A Molecular 3 Docking Approach' described a computational study to search for potential plant-based natural products as potential Enterococcus faecalis SrtA inhibitors with the aim of treating endodontic infections. The paper is generally well-written. However, the authors need to address the following points before publication:

1. Page 1-2, Introduction: In the abstract section, the authors mentioned that they used Ampicillin as the control drug in the study. However, an introduction to Ampicillin's effect on SrtA proteins, is missing.

2. Page 2, Introduction, Paragraph 6, Line 88: It would be better to explain the advantages of computational studies versus X-ray crystal studies on this target.

3. Page 3, Section 2.2: The authors need to explain the rationale for selecting the control compound. This could include biochemical assays, crystal structure, or other evidence that proves Ampicillin is a binder to SrtA proteins.

4. Page 4, Table 2: The authors did not describe how the table is arranged. It is neither arranged by the name of the compound nor by the value of binding energy.

5. Page 4-5, Table 2: Both Glabridin and Pinocembrin are listed twice in the table.

6. Page 6, Paragraph 1, Line 153: There is a typo in 'Figure. 3.'

7. Page 14, Discussion, Paragraph 1, Line 393: '... Ursolic acid demonstrated unique binding patterns.' The authors did not disclose any detailed results of computational studies on Ursolic acid other than the binding energy and amino acid residue interaction listed in Table 2. The authors should either disclose more details on their results regarding Ursolic acid or remove the mention of 'Ursolic acid' from this sentence.

8. Page 16: Remove section 16 unless the authors introduce related information here.

Author Response

Exploring Plant-Based Compounds as Alternatives for Targeting Enterococcus faecalis in Endodontic Therapy: A Molecular Docking Approach

Manuscript ID: ijms-3073202

We thank all the reviewers and acknowledge the comments made by all the reviewers.

Answers to reviewer 3 comments

Query 1:  Page 1-2, Introduction: In the abstract section, the authors mentioned that they used Ampicillin as the control drug in the study. However, an introduction to Ampicillin's effect on SrtA proteins, is missing.

Answer 1: Thank you and we acknowledge the comment. We have included the details regarding Ampicillin usage in the Materials and methods under 4.3 Ligand identification and selection section of the revised manuscript for better clarity and understanding.

Query 2: Page 2, Introduction, Paragraph 6, Line 88: It would be better to explain the advantages of computational studies versus X-ray crystal studies on this target.

Answer 2: Thank you and we acknowledge the comment. As suggested by the reviewer we included the advantages of computational studies in the introduction of the revised manuscript.

Query 3: Page 3, Section 2.2: The authors need to explain the rationale for selecting the control compound. This could include biochemical assays, crystal structure, or other evidence that proves Ampicillin is a binder to SrtA proteins.

Answer 3: Thank you and we acknowledge the comment. As suggested by the reviewer, we included the details under the Materials and Methods section of the manuscript.

Query 4: Page 4, Table 2: The authors did not describe how the table is arranged. It is neither arranged by the name of the compound nor by the value of binding energy.

Answer 4: We acknowledge the comment. As suggested by the reviewer, the compounds in Table 2 are ordered based on the binding energy.

Query 5: Page 4-5, Table 2: Both Glabridin and Pinocembrin are listed twice in the table.

Answer 5: We acknowledge the comment. The contents of Table 2 is corrected.

Query 6: Page 6, Paragraph 1, Line 153: There is a typo in 'Figure. 3.'

Answer 6: We acknowledge the comment. The typo is corrected.

Query 7: Page 14, Discussion, Paragraph 1, Line 393: '... Ursolic acid demonstrated unique binding patterns.' The authors did not disclose any detailed results of computational studies on Ursolic acid other than the binding energy and amino acid residue interaction listed in Table 2. The authors should either disclose more details on their results regarding Ursolic acid or remove the mention of 'Ursolic acid' from this sentence.

Answer 7: We acknowledge the comment. As suggested by the reviewer, the sentence is removed. 

Query 8: Page 16: Remove section 16 unless the authors introduce related information here.

Answer 8: We acknowledge the comment. As suggested by the reviewer, the section is removed.

Round 2

Reviewer 2 Report

Comments and Suggestions for Authors

Thanks for making the required corrections. But why the Hydrophobic & Hydrogen interaction between protein and ligands in Figure 2 and in table 2 are not matching? Please explain or provide a more clear image.

Author Response

Exploring Plant-Based Compounds as Alternatives for Targeting Enterococcus faecalis in Endodontic Therapy: A Molecular Docking Approach

Manuscript ID: ijms-3073202

We thank all the reviewers and acknowledge the comments made by all the reviewers.

Answers to reviewer comment

Query 1: But why the Hydrophobic & Hydrogen interaction between protein and ligands in Figure 2 and in table 2 are not matching? Please explain or provide a more clear image.

Answer 1: Thank you and we acknowledge the comment. We checked the Figure 2 and Table 2. We verified the interactions again and found missing residue labels. We now added the correct labels (Val147 in pinocembrin complex and Val88 in ampicillin complex) in Figure 2. In addition, we also corrected the Figure 2 captions.
